# A Deep-Learning Based Method for Analysis of Students' Attention in Offline Class

**Xufeng Ling** [1], **Jie Yang** [2,*], **Jingxin Liang** [1], **Huaizhong Zhu** [1,*] **and Hui Sun** [3]

1 AI School, Tianhua College, Shanghai Normal University, No. 1661 North Shengxin Road, Shanghai 200234, China
2 Institute of Image Processing and Pattern Recognition, Shanghai Jiaotong University, No. 800 Dongchuan Road, Shanghai 200240, China
3 Shanghai Technical Institute of Electronics & Information, No. 3098 Wahong Road, Fengxian District, Shanghai 201411, China
* Correspondence: jieyang@sjtu.edu.cn (J.Y.); zhz2282@sthu.edu.cn (H.Z.)

**Abstract:** Students' actual learning engagement in class, which we call learning attention, is a major indicator used to measure learning outcomes. Obtaining and analyzing students' attention accurately in offline classes is important empirical research that can improve teachers' teaching methods. This paper proposes a method to obtain and measure students' attention in class by applying a variety of deep-learning models and initiatively divides a whole class into a series of time durations, which are categorized into four states: lecturing, interaction, practice, and transcription. After video and audio information is taken with Internet of Things (IoT) technology in class, Retinaface and the Vision Transformer (ViT) model is used to detect faces and extract students' head-pose parameters. Automatic speech recognition (ASR) models are used to divide a class into a series of four states. Combining the class-state sequence and each student's head-pose parameters, the learning attention of each student can be accurately calculated. Finally, individual and statistical learning attention analyses are conducted that can help teachers to improve their teaching methods. This method shows potential application value and can be deployed in schools and applied in different smart education programs.

**Keywords:** smart education; smart class; learning attention; deep learning; head-pose estimation

## 1. Introduction

Student's learning attention in an offline class is a quantitative index to measure students' learning engagement in class, which includes attention depth and attention duration. The deeper and longer students' attention is in the class, the better the learning results will be. The famous Russian educator Uscinski formerly said that "learning attentiveness is the only gateway to our soul, and everything in consciousness must pass through it to enter". There is a great similarity between attentive learning and immersive learning. A high learning attention state easily leads to an immersive learning state. Studies show that the learning outcomes of immersive learning can reach five times that of ordinary learning. Learning attention is related to a learner's learning motivation and learning interests. Strong learning objectives and intensive learning interests, such as happy learning or inquiry learning, can easily drive learners to enter a high-focus learning state. Well and carefully designed teaching methods can stimulate students' learning interest in class and improve their learning attention.

The characteristics of learning attention vary among students and depend on the special characteristics of each student [1,2]. One student may easily enter a high learning attention state but cannot keep it for long. Another student may find it difficult to enter the state of high attention, but once entered, they can maintain this state for a long time. Some students struggle to stay engaged in their studies and are easily affected by external factors. We believe that studying the students' whole learning attention in class can measure the

teaching effect of the teacher and make a quantitative index to measure the effect of class teaching. Students' learning attention can also help teachers to improve their teaching methods. Studying the personal learning attention of each student can help customize personalized learning programs for each student.

The research on online learning attention has attracted researchers and achieved valuable results. In practice, the degree of students' class attention is difficult to measure. It is a challenging job to accurately detect the learning status of students in the class. Some scholars have conducted research [3,4] on how to detect learning attention and achieved valuable results. Sun et al. [5] reviewed the existing progress in learning attention at home and abroad, divided the focus research into two aspects, attention recognition based on facial expression and attention recognition based on behavior, and further discussed the development trend of attention recognition. Focus research scenarios mainly include the online education environment and offline class teaching environments.

For the convenience of image and voice acquisition in online classes, researchers have studied attention recognition in online classes and made good progress. Wang et al. [6] proposed a method of building a structural equation model based on the Triadic Theory of Learning, which studies the efficiency of online learning, divides the teaching quality into deep learning and shallow learning states, and tests the online teaching quality by taking 636 students majoring in economics and management in national universities as samples. This method puts forward that learning attention heavily affected students' learning vigor and was heavily affected by teachers' teaching contents and teaching method, further optimizing the configuration of the online teaching platform to improve the efficiency of online teaching. Chang et al. [7] proposed a method that consists of 3 sub-modules. The first module is for head pose detection, which mainly detects the deviation angle of the head of each student. The second module scores fatigue by eye and mouth closure. The third module detects facial expressions and scores emotion. Then, by merging the above information, a fuzzy comprehensive evaluation method is used to quantitatively evaluate their learning attention. The learning attention detection system of the online education platform designed by this algorithm has been tested in a simulated scene. The experiment shows that this method can effectively evaluate students' class focus and improve the class quality and students' learning outcomes. Deng et al. [8] proposed a machine learning-based approach to measuring students' learning attention. A Gabor wavelet technique is used to extract features of eye states, and a support-vector machine (SVM) model is trained to classify students' eye states. Experiments show that the methods have good performance and have value in real applications. Zhong et al. [9] proposed the analysis model of college students' class behavior based on deep learning technology and constructed an analysis system for the college students' class behavior to realize the two core functions of learning attention analysis, providing an intelligent and efficient way of College Students' behaviors analysis and supervision and supporting college students' behavior research and management.

Currently, IoT technologies and deep learning methods are used in smart classrooms, continuously improving teaching methods and obtaining more accurate evaluation results [10–12]. To test the effectiveness of a machine vision-based approach, Goldberg et al. [13] proposed a new validated manual rating method and provided a method for a machine vision-based approach to evaluate students learning attention. The experiment results show that the manual rating system was significantly correlated with self-reported study engagement. Zaletelj et al. [14] proposed a novel approach to automatically estimate students' learning attention in the classroom. A Kinect One sensor is used to acquire 2D and 3D data and build a feature set of both facial and body properties of students. Machine learning algorithms are used to estimate the time-varying attention levels of each student. The experiment results show that this method can detect students' attention and average attention levels, and the system has potential practical application for non-intrusive analysis of the student learning attention. Leelavathy et al. [15] proposed a novelty method to use biometric features such as eye gaze movements, head movements and facial emotion to detect students' study

engagement with many techniques, including principal component analysis (PCA), Haar cascade, local binary patterns and OpenCV, which are used for facial emotion recognition, pupil detection, head movements recognition and machine learning. Ling et al. [16] proposed a facial expression recognition and a head pose estimation system for smart learning in the classroom by using the YOLO model to detect student faces in high-resolution video and using a self-attention-based ViT model to recognize facial expressions. The classified facial expression is used to assist the teacher in analyzing students' learning status to provide suggestions for improving the teaching effect. As the traditional offline class is still the most important teaching place, some researchers have focused on the student's attention in the traditional class. He et al. [17] proposed a learning expression automatic recognition method integrating local and global features, which extracts and integrates the local geometric features of an expression image; it uses CLBP (complete local binary pattern) global shallow texture features reduced by KPCA (kernel principal component analysis) and CNN global depth network features. In addition, a new spontaneous learning expression database is also constructed, which divides the emotions in classroom learning into five types: confusion, happiness, fatigue, surprise, and neutrality, which are used for the training of CNN model. Comparative experiments show that this method is not only better than the traditional facial expression recognition method but effectively obtains the emotional changes of middle school students in the classroom, helps teachers grasp the overall situation of class students accurately and comprehensively, and promotes the improvement of classroom teaching quality. Sun [18] proposed a method that takes students' heads up and down as a method to identify learning attention. The situation of students' head position in the classroom, whether up or down, is detected every 50 frames. Combined with the comprehensive consideration of students' grades, Sun studied the difference in students' attention and the distribution of time periods with high attention in the classroom and obtained a positive correlation between attention and grades. It was also concluded that the peak attention of each class is in the first 10 min, 21–30 min, and 5 min before the end of the class. Teachers can teach students at different levels according to the above data to improve the effectiveness of learning. Duan et al. [19] proposed a method to detect the learning attention of each student by eye-opening and closing based on raising and lowering the head. When the experimental sample was 60 people in a class, the accuracy of this algorithm reached 92%, which improves the accuracy of recognition compared with the traditional learning attention algorithm. Yin et al. [20] proposed an intelligent teacher management system that can improve students' learning efficiency and make students more focused on learning. The system can monitor the state of students through wireless sensors. When the students are not engaged in a learning state, the system can stimulate the enthusiasm of students. At the same time, the state of students and the state of the class will be recorded in the internal background system. The system managers can monitor the state of students and teachers in a timely fashion and give some guidance and feedback to students and teachers.

Scholars have applied new technologies in class to improve learning effects and have obtained good results [21,22]. We believe that traditional offline class teaching has irreplaceable advantages such as face-to-face communication, a unified learning environment and an immersive learning scene. It is of great significance to detect students' learning attention in traditional offline classes. Compared with online education, we use cameras and recording equipment to collect video and audio in the traditional class. The taken video is easily disturbed by different light conditions, ambient noise and so on. We believe that the use of eye-opening and closing detection, facial expression analysis, fatigue analysis [23] and other methods needs higher quality video and audio that the traditional class cannot provide. It is difficult to collect such high-quality videos in practice, and the deep learning model also faces difficulty in analyzing such high-resolution images. Due to the limitations of data collection equipment in traditional classrooms, this paper proposes a deep learning method based on the self-attention mechanism. According to the traditional class teaching scene, the students' head pose parameters combined with the class states can identify the

students' learning attention. There are face-detecting tools in our toolbox to detect faces. There is a head pose recognition model for obtaining head pose parameters. With speech recognition tools, a class can be divided into lecture, interaction, practice, and transcription states. Finally, we use multi-modal analysis methods to analyze the rise of students in different class states to achieve a more accurate analysis of students' attention. The proposed method in this paper is based on the following assumptions. (1) When a teacher is giving a lecture or interacting with students, such as asking a question and waiting for the students to answer, we think that students should look at the teacher. If the students are not looking at the teacher, but lowering their heads or faces to the left or right, then we do not think they are focused. (2) When the class is in the practice state, students should lower their heads and write; otherwise, we do not think they are focused. (3) When a student is in the state of taking notes, he should look up at the blackboard and then lower his head to copy. If he is not in this state, we also think he is not in the state of focus.

The key contributions of the proposed method include the following.

- Different from many existing studies on online student learning attention, our study aims to analyze student learning attention in a traditional offline class. The input data, such as face images, face expressions, and voices, are relatively difficult to acquire due to occlusion, noises and light conditions and other factors.
- Different from the traditional questionnaire survey method, this paper uses the observation and analysis method based on artificial intelligence to carry out the research on smart education. It collects videos and sounds in class through Internet of Things (IoT) technology and uses machines instead of people to observe students. It can capture the most real scene in class and uses artificial intelligence technology to compute and analyze students' learning attention.
- This paper initially divides a class into multiple time periods, each of which can be categorized into four states of lecturing, interaction, practice, and transcription and obtains the differences in students' learning attention features in the four different class states.
- Since the image acquisition method in the offline class is limited, the image patch of students' faces in the video is small enough that we cannot analyze face expressions to obtain student learning attention. We use many deep learning models, such as Retinaface [24], ViT [25–27], and ASR [28–30], for face detection and location, head pose estimation [31,32], and speech recognition to accurately extract the learning attention of each student.
- This paper analyzed each student's learning attention and carried out a total statistical analysis so that we can help each student improve his/her learning outcome and help teachers improve their teaching effects.
- The proposed method has a high application value and can be deployed in classrooms or laboratories in high schools and universities.

## 2. Method

The first step is to use Retinaface for face detection. Since the complex background of the image will affect the detection result of Retinaface, we carefully set the background to a relatively simple set. The Retinaface model needs no training, the results of face detection are good, and the speed is fast. The second step is to select Vision Transformer as the head pose regression model. We train a ViT-based network using the training dataset and estimate the students' head pose after the network converges; the network outputs two angle parameters, yaw and pitch. The third step is to correct the head pose parameters. In practical use, due to the influence of the camera orientation and the teacher's moving position in the classroom, we should correct the head pose parameters to obtain the correct student head pose. The fourth step is to use a sound type clarification model and a speech recognition model to identify many different states in a class. The fifth step is multi-modal analysis, which can merge the state information of the class with the head pose parameters

of students and find in the current class state if the specific head pose of each student is engaged in learning.

The system is composed of two branches, as shown in Figure 1. The left branch is video processing, mainly including video acquisition, face detection, head pose estimation, and two-phase correction, such that the head pose parameters of each student are obtained. The branch on the right includes sound processing, including audio acquisition, audio classification, speech recognition, speech duration calculation, and finally, identifying the class state. By merging the results of the left branch and the right branch, we can then analyze the learning attention of each student in the class.

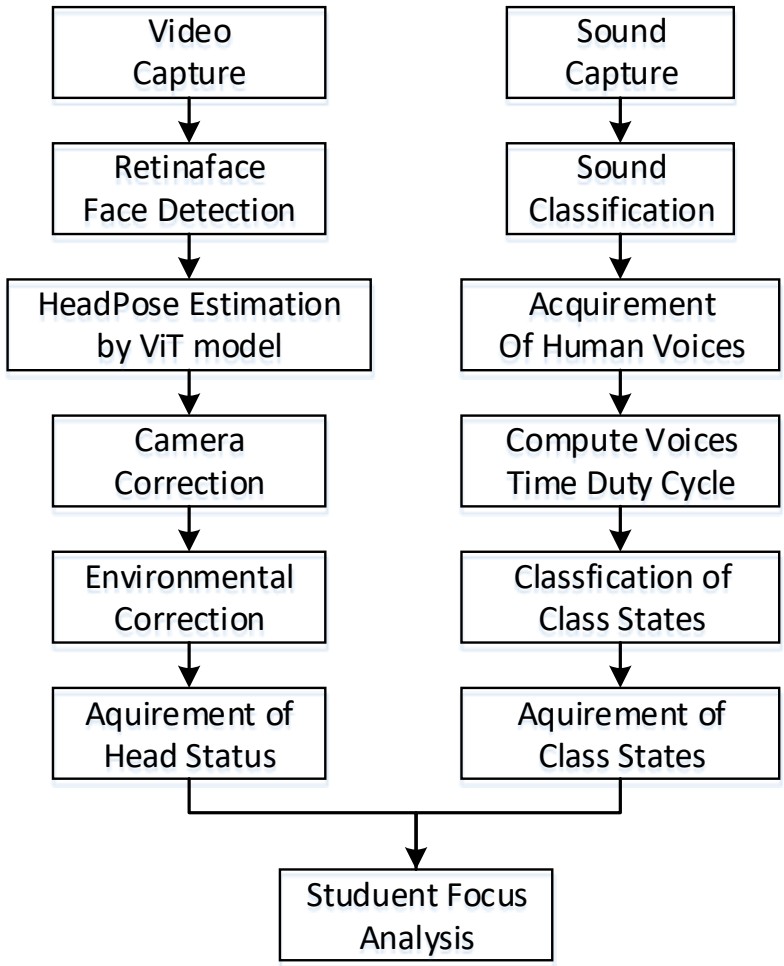

**Figure 1.** Flowchart of the system.

### 2.1. Introduction of Retinaface

Retinaface is a face detection model proposed by the Insightface team in [24], which can improve the human face detection accuracy of Arcface from 98.37% to 99.49%, showing that it has reached the face detection of the front face and side face. The model is open-source by Insightface and was built using the MXNet framework. Retinaface improves on Retinanet based on a detection network, adds the three-level joint detection model of the SSH network, and improves the detection accuracy. Figure 2 shows the result of Retinaface. The open-source community provides two versions of the model (MobileNet and ResNet). MobileNet version is a lightweight model which can provide faster detection speed with less computing power. The ResNet version is relatively larger, thus requiring higher computing power and higher accuracy. The model in this paper uses two versions at the same time. The portable system uses the lightweight model, and the offline system uses the ResNet version.

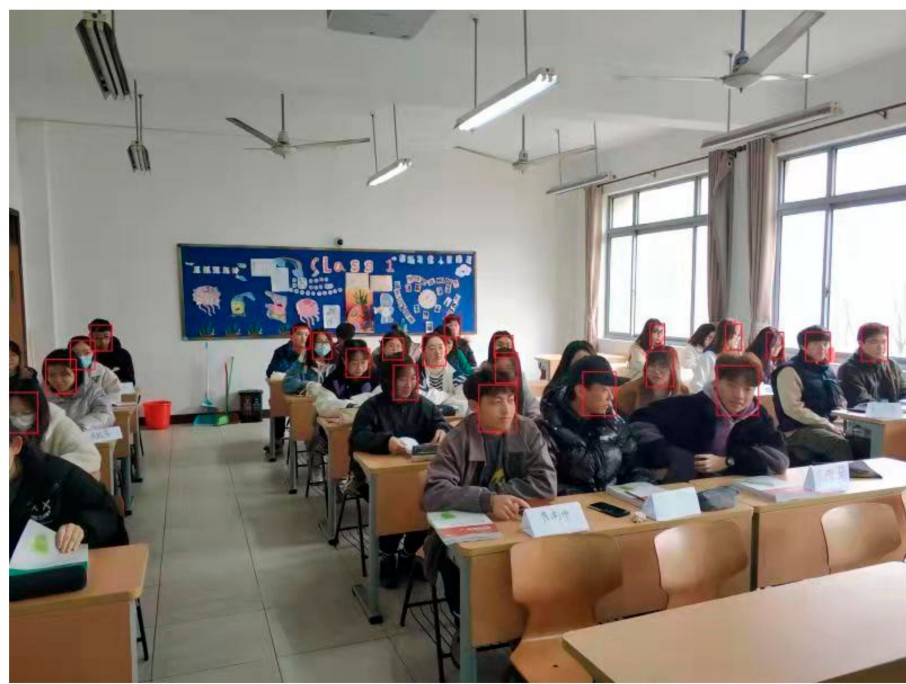

**Figure 2.** The result of Retinaface for the detection of faces in the classroom.

### 2.2. Vision Transformer

This job of head pose regression is to estimate the 3-dimensional Euler angles, yaw, pitch and roll, of the head of each student in the 2-dimensional images taken in the classroom by setting the coordinate system of the camera as the reference coordinate system. As shown in Figure 3, yaw, pitch and roll represent three different directions of the face angle. Head pose estimation is a difficult job and is related to the 3-dimensional modeling of the human head. We use a labeled head pose dataset to train a vision transformer regression network, and then the trained network predicts the head pose. For a given face image, the method in this paper can directly predict the three Euler angles of head poses and use the MSE loss function to train the vision transformer network.

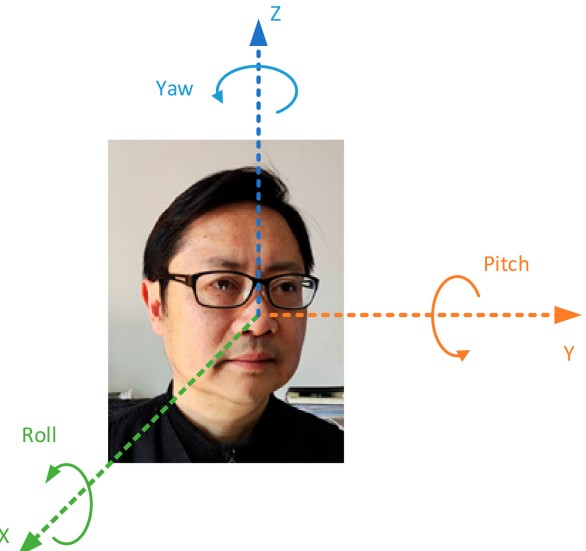

**Figure 3.** Three Euler angles of head poses.

### 2.2.1. Self-Attention

A transformer is a deep learning model based on self-attention [23]. Compared with CNN, RNN and other models, a transformer has two significant advantages. One is that it can learn in the way of self-supervised learning, and the other is that the model structure supports parallel computing. The BERT model based on the transformer mechanism has achieved great success in the field of natural language processing. The basic units of a transformer are an encoder and a decoder. Each encoder is composed of a self-attention layer and an MLP composed of $Q, K,$ and $V$. The decoder is composed of an attention layer, an encoding decoding attention layer and an MLP. Multiple codecs and decoders are connected in series to form a transformer.

The main function of the self-attention layer is to model the relationship between each input value of the input sequence. Let the input sequence $= \{x_1, x_2, \ldots, x_n\}$ and the dimension be D. after learning and synthesizing the overall information of the input sequence, we learn the relationship between other values in the sequence from the attention layer.

$$Q = W_q \times X; \; K = W_k \times X; \; V = W_v \times X \tag{1}$$

We set the query as $Q$, the index as $K$ and the value as $V$; then, the input sequence $X$ is mapped through the matrices $W_q, W_k, W_v$ to obtain $Q, K,$ and $V$, as shown in formula (1). The above $W_q, W_k,$ and $W_v$ are acquired by training.

$$Z = \text{softmax}\left(\frac{QK^T}{\sqrt{d_q}}\right)V \tag{2}$$

We calculate the $Z$ value of attention through Formula (2). The multi-head-attention mechanism is introduced. Each head calculates the association relationship between two different position elements of the input sequence. Multiple-head attention can calculate multiple relationships, generally including 6 heads, 8 heads and 12 heads, as shown in Figure 4.

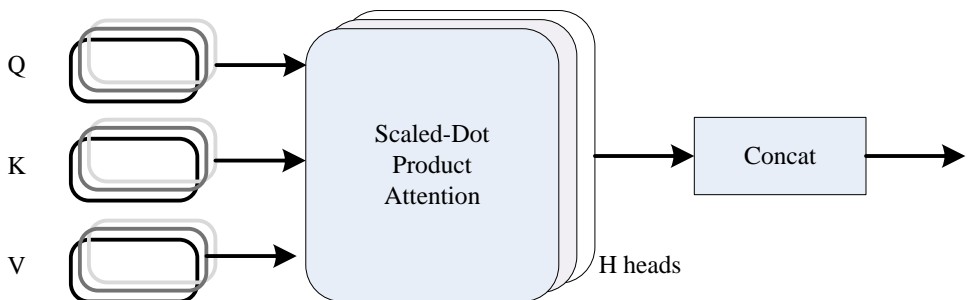

**Figure 4.** Multi-head self-attention.

### 2.2.2. ViT Model

The vision transformer improves the original transformer so that the model can support the embedding of 2D images. Assuming that the image is composed of a lattice of $W \times H$ pixels, the image is first divided into image blocks of multiple $N \times N$ pixels. The quantitative dimension of the image block ($\frac{W}{N} \times \frac{H}{N}$), $N$, is generally 16 or 32, and both $W$ and $H$ can be divided by $N$. These image blocks are connected in series to form a sequence of $L = \frac{W}{N} \times \frac{H}{N}$ image blocks. Through the embedding module, the image block is converted into a d-dimensional vector ($D$ is generally 384, 512, 768 or 1024). Image embedding can realize the coding work of compressing redundant information and extracting features. We add the CLS flag bit (used for classification or regression representing global information) at the first place of the embedded sequence to obtain an $L + 1$ vector patch_ embedding. In order to maintain the position information, the position embedding vector POS is constructed embedding dimension $L \times D$. Different

from the traditional transformer, the position embedding vector in the ViT model is not constructed manually but is generated through training embedding and POS embedding. Two vectors are added to form self-attention. The vision transformer regression model is shown in Figure 5.

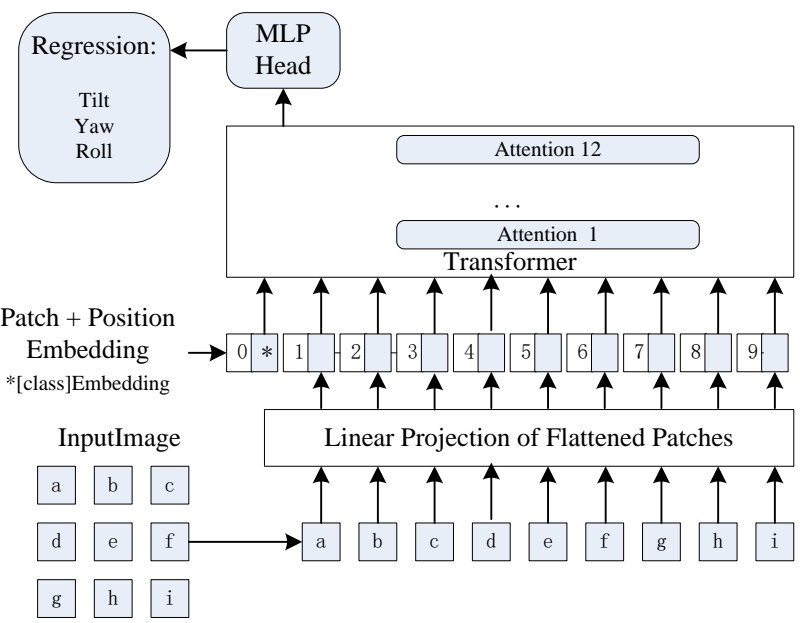

**Figure 5.** Vision Transformer. a, b, c, ... , i are image patches.

The input information of each attention layer can be extracted from the initial attention vector and the self-correlation of each attention layer; each attention layer includes initial calculation, regularization, residual error, feed-forward, residual error, feed-forward and other steps of self-attention, including the input information of the global attention vector and the self-correlation of each attention layer. Finally, the ViT model outputs CLS bits and maps the D dimension to the N dimension through an MLP. If it is a classification task, N is the number of categories; if it is a regression task, N is the dimension of the regression task.

The output in this paper is two-dimensional data, namely yaw and pitch. The error function adopts MESLoss, and the optimization function adopts Adam. In the process of network training, the error is calculated through the above steps, and then the network is trained backward. In the process of reasoning, the test samples are input into the trained network, and the output value is the estimation of regression parameters.

The number on the left in the bracket is the yaw angle, and the number on the right in the bracket is the pitch angle, as shown in Figure 6.

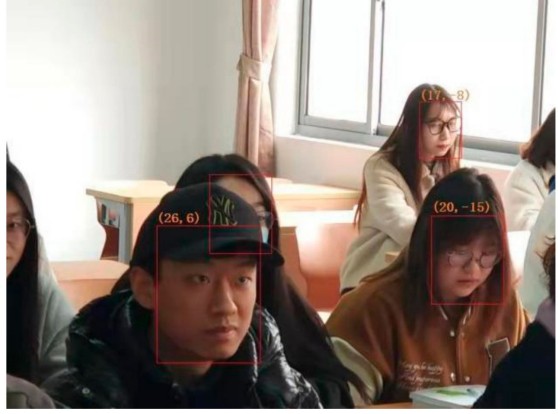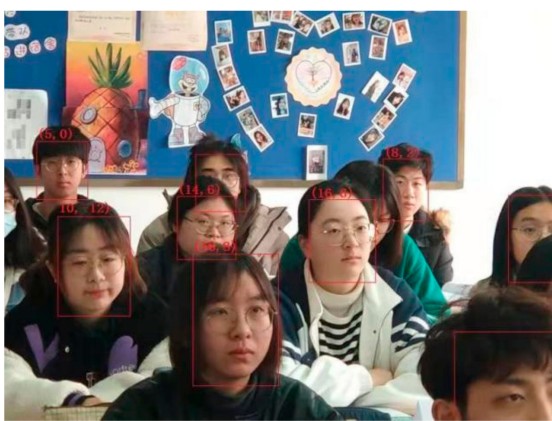

**Figure 6.** Head-pose regression.

*2.3. Parameter Correction*

2.3.1. Phase One Correction

　　We set all the students so that they are sitting well in their positions and all positions are occupied. All students look straight ahead and take a standardized picture. Then the photos are input into the trained ViT model, which outputs the head posture parameters of students at different positions and records the abscissa and ordinate of the position of each student's head at the same time. This head pose parameter is called the preset correction parameter.

　　In the actual classroom data collection, we input the trained ViT model and output the head pose parameters for each picture. Meanwhile, we calculate the coordinates of the face, find the nearest preset coordinates, take out the preset parameters of the modified coordinate head pose, and then subtract the current head pose parameters from the preset calibration parameters to complete a correction, as shown in Formula (3).

$$\theta_n = \theta - \theta_0, \ \varphi_n = \varphi - \varphi_0 \tag{3}$$

2.3.2. Phase Two Correction

　　In the actual calculation, we found that the students' perspective will move with the teacher. When the teacher teaches, s/he will walk around in front of the podium, and the students' head posture will change accordingly. This is the average head pose of all students by calculation. By obtaining the mean value of yaw and pitch and using the mean value to correct the face angle of each student twice, we attain more accurate head pose parameters. The subsequent attention analysis is carried out in the head pose parameters after phase two correction, as shown in Formula (4).

$$\theta_n = \theta - \theta_0 - \theta_{avg}, \ \varphi_n = \varphi - \varphi_0 - \varphi_{avg} \tag{4}$$

*2.4. Classification of Class State by Voice Recognition Method*

　　According to the teaching methods seen in traditional classes, we divide the class state into four states: lecturing, interaction, practice, and transcription. The characteristic of the lecturing state is that the teacher is talking all the time. The system can detect that the teacher continues to output voice. The voice duty cycle is generally greater than 50%, and only the teacher is talking. The interaction state is the interactive state between teachers and students; that is, the teacher asks questions, and students answer. The system can confirm that state by detecting the voices of both the teacher and students. The voice duty cycle is generally greater than 50%. In the practice state, after the teacher arranges the classroom practice, the students start to practice in the class. The characteristics of this state are that the system cannot detect the voice, or the voice cannot be detected most of the time, and the voice duty cycle is less than 10%. In the transcription state, the teacher writes on the blackboard while speaking, and students are doing transcription. The characteristics of this state are that the system detects that the voice duty cycle is greater than 10% and less than 30%, and the teacher is speaking. The processing process is shown in Figure 1.

　　For environmental sound classification, we use a convolutional neural network CNN named ECAPA-TDNN [33] to train a model that can recognizes environmental sounds. The model can distinguish voice, animal sounds, indoor sounds, outdoor sounds, and human action sounds. By using this model, we can distinguish the speech from the environmental sound. The neural network is shown in Figure 7.

　　In Figure 7a, k is the kernel size for dilation spacing of the Conv1D layers or SE-Res2Blocks. *C* and *T* correspond to the channel and temporal dimension of the intermediate feature maps, respectively. *S* is the number of categories. We have used the Urbansound8k dataset to train the ECAPA-TDNN model. Urbansound8k is a public dataset widely used for automatic urban environmental sound classification research and includes 10 category sounds: air conditioners, sirens, children playing, dog barks, jackhammers, engines idling, gunshots, drilling, car horns, and street music. We remove the children playing sound

samples and add human voice samples to form a new dataset to train our specific model. The trained model can recognize human voices from environmental sounds accurately, and the recognition accuracy is higher than 98.6%. The confusion matrix is shown in Figure 8.

After obtaining speech information, we use ECAPA-TDNN to train a voiceprint recognition model that can recognize different people's speech. The model can distinguish whether the teacher is speaking or students are speaking, as well as whether multiple people are speaking or one person is speaking.

For voice duty cycle computation, we know that a class is 30 min in length. We divide each class into 1-min units in a sequence with a length of 30. Each unit uses the model for classification. After the voice is detected, the voice duration can be obtained, and then the voice duration duty cycle can be calculated.

For class state recognition, each unit uses the model to identify whether a human voice is speaking, and then the voiceprint model is used to detect whether there are multiple people's voices. Therefore, the following classification process is performed. If the voice duty cycle is less than 10%, the state is determined to be the practice state. If the voice duty cycle is greater than 10% and less than 30%, the state is determined to be the transcription state. If the voice duty cycle is greater than 50% and multiple people speak, the state is determined to be interactive, and if the voice duty cycle is greater than 50% and a single person speaks, the state is determined to be teaching.

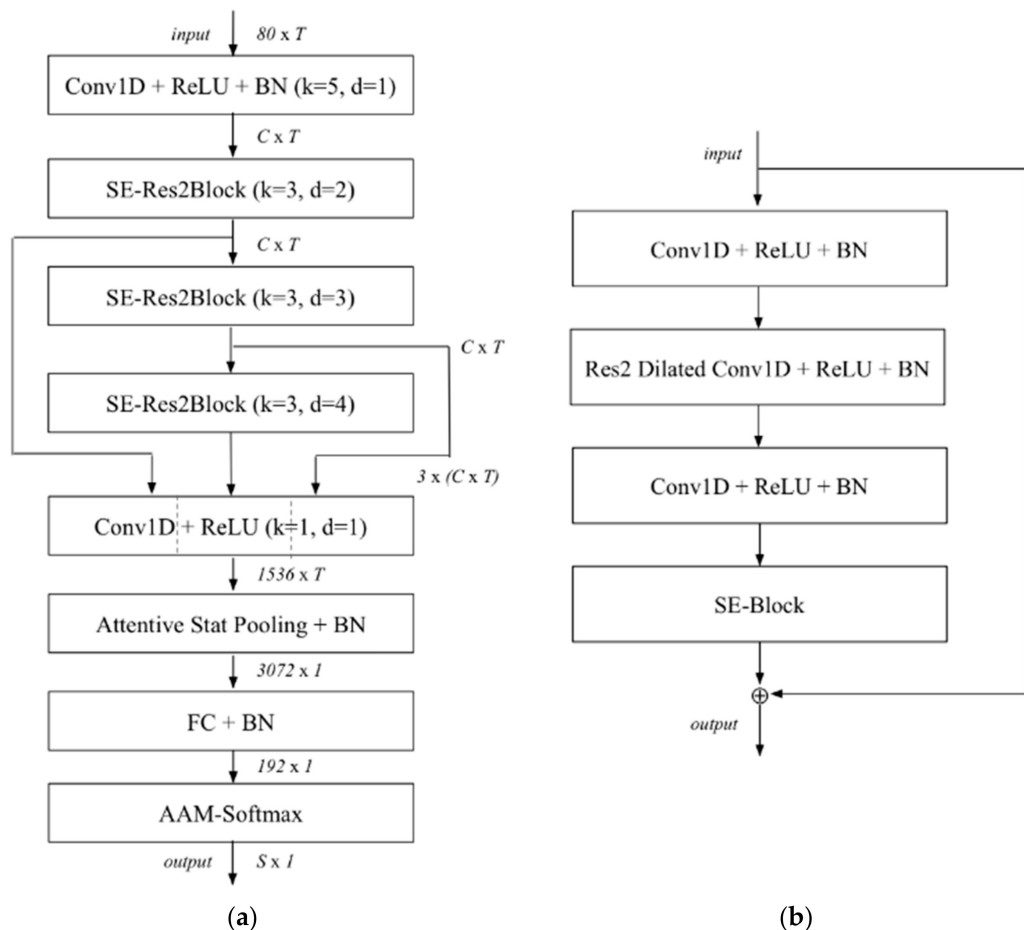

**Figure 7.** The network architecture of ECAPA-TDNN. (**a**) Network topology of ECAPA-TDNN. (**b**) The SE-Res2Block architecture.

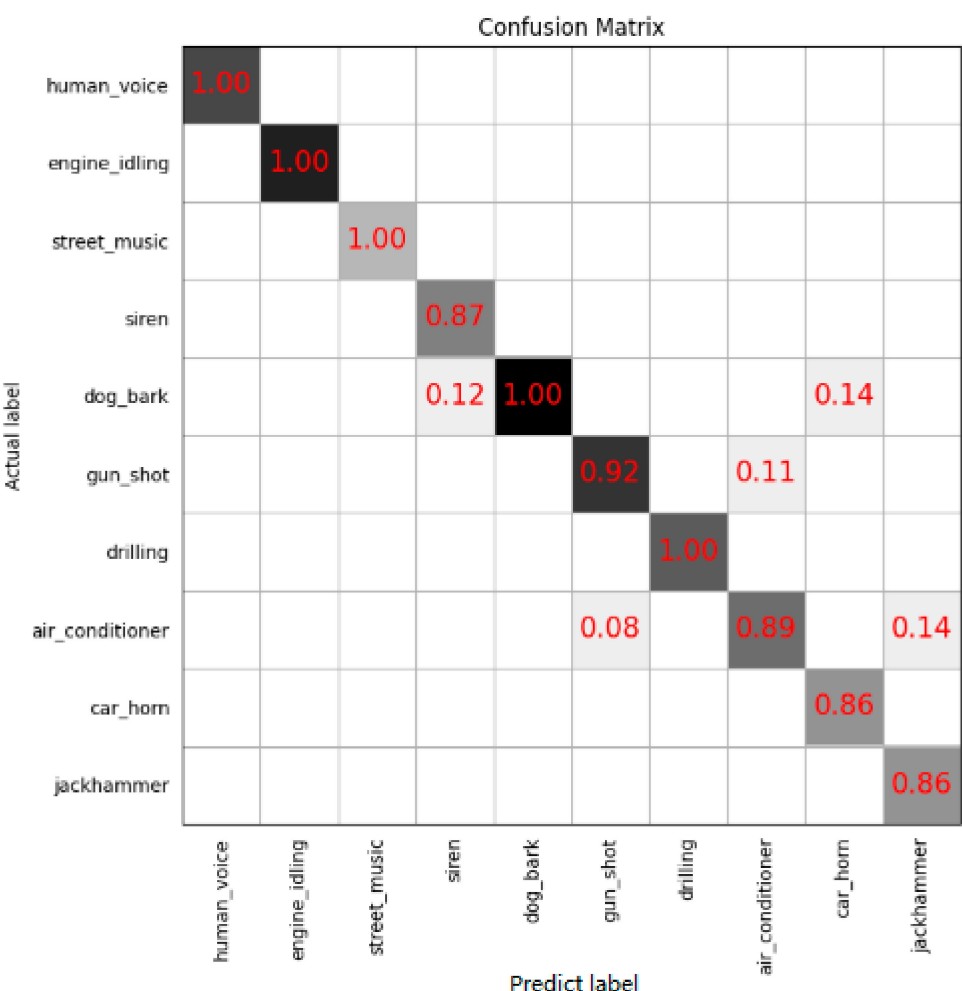

**Figure 8.** The confusion matrix of the classification results after 20 epochs.

*2.5. Analysis of Students' Individual and Class Attention*

After obtaining the head pose parameters of students, we can classify students into forward-looking states, head-down states and head-side states. If a student bows his head and the angle exceeds 15 degrees, then we call it a head-down state. Additionally, if a student turns his head left or right and the angle exceeds 20 degrees, then we call it a head-side state. Otherwise, we call it a forward-looking state, as shown in Table 1. The head pose state of a student can indicate if the student is listening to the teacher or if their mind is absent.

**Table 1.** Head pose parameters and student state.

| Forward-Looking State | Head-Down State | Head-Side State |
|:---:|:---:|:---:|
| Pitch $\geq -15$ | Pitch $< -15$ | Yaw $\geq 20$, or yaw $< -20$ |

In addition, according to the four different states of the traditional offline class, the performance of students' head poses and learning attention is different, as shown in Table 2.

We need different sampling frequencies for classroom analysis according to different class states. For example, when teachers are lecturing, we only need to collect an image every minute because students are generally in a state of having their heads raised. In practice, we only need to collect an image each minute because students usually lower their heads and wait after completing the practice. In the state of teacher-student interaction, students are generally in the state of having their heads raised. However, in the state of

transcription, we must increase the interview rate because the characteristic of attention at this time is that students quickly raise their heads towards the blackboard and then bow their heads before writing. Thus, the video sampling rate needs to reach 2 frames per second.

**Table 2.** Class state, student learning attention and head pose state.

|  | **Focused Attention** | **Unfocused Attention** |
|---|---|---|
| Lecture | Forward-looking state | Head-down state<br>Head-side state |
| Interaction | Forward-looking state | Head-down state<br>Head-side state |
| Practice | Head-down state, or head-down state, then forward-looking state | Forward-looking state<br>Head-side state |
| Transcription | Forward-looking state for a short time, then head-down state to copy, repeatedly | Head-side state |

According to the classroom state, different sampling frequencies are used to record a focus state sequence for each student, and the learning attention of each student is analyzed by this state sequence. According to the analysis of students' personal learning attention, each student has their own unique personality. Some students can easily enter the state of learning attention and maintain attention for a long time. Such students can enter an immersive learning state, which has high learning outcomes. Therefore, we should calculate each student's learning attention, average attention time and maximum attention time.

At the same time, when we generate the learning attention sequence of all students, we can do further statistical analysis, especially the horizontal synchronous cross-sectional analysis, such as the overall analysis of learning attention, the average duration of class attention, etc., which generates an attention matrix for each class and provides a basis for analyzing effects.

## 3. Experiment and Analysis

The experimental work is divided into two parts: model training and model inference, which are completed on different software and hardware platforms. The training environment used a Linux server, configured with 128 G memory, four RTX 2080Ti GPU computation cards, and the Ubuntu 20.04 operating system. The development platform for the video was the mainstream deep learning environment Pytorch 1.7.1, and the development platform for sound was PaddlePaddle, with the Python programming language. The development environment used minconda3 to create an independent experimental environment. The portable inference system adopted an NVIDIA Xavier AgX 32 g high-performance edge terminal, installed with an Ubuntu 18.04 system, minicanda3, Pytoch 1.7.1 and PaddlePaddle. An 1080p high-resolution industrial camera and high-sensitivity pickup head were used to collect video and audio signals. The offline system was equipped with a workstation consisting of 2 RTX 2080Ti GPU computation cards, the Ubuntu 20.04 operating system, and a Pytorch 1.7.1 environment. We selected a public dataset named Head Pose Database as our dataset. All images were taken using the FAME Platform of the PRIMA Team in INRIA Rhone-Alpes. The pitch angle ranged from $-90°$ to $+90°$ and the yaw angles ranged from $-60°$ to $+60°$ across 2790 images.

The study design is as follows. There are always cameras on the wall in front of the classroom. The students do not know whether these cameras are collecting data, or a few of them know these cameras are collecting data in class, but they do not know if the collected data are used to analyze their learning attention. In this case, it can be considered that the data collected by the cameras will not affect students' learning behaviors. We used the cameras to collect and analyze the students' learning attention of teachers and made a comparison of them. We also have collected different courses, some of which are

theoretical courses such as mathematics or programming language grammar teaching, and some of which are interactive and practical courses such as social hotspot case analysis or programming project implementation.

### 3.1. Training and Classification of ViT

We conducted head pose data preprocessing, ViT model training, model verification and model testing. This paper used the public dataset Head Pose Database, which was taken by PRIMA Team0 of INRIA Rhone Alpes [29]. The head pose dataset collected the head pose images of 15 people. The yaw angle ranged from −90° to +90°, and the pitch angle ranged from −60° to +60 across 2790 images. For each person, 2 series of 93 pictures (93 different poses) are available. The samples in the dataset have different skin colors. The background is clear and convenient for facial image operation. Samples of the dataset are shown in Figure 9.

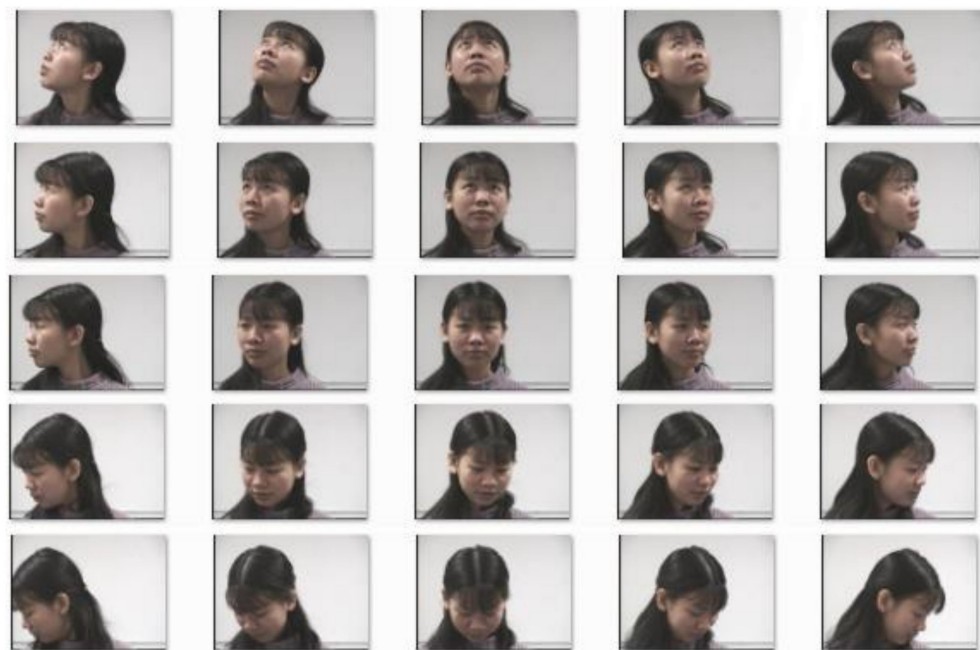

**Figure 9.** Head pose image samples.

Each person has 13 head yaw angles: −90°, −75°, −60°, −45°, −30°, −15°, 0°, +15°, +30°, +45°, +60°, +75°, and +90°. Similarly, there are 9 head pitch angles: −60°, −45°, −30°, −15°, 0°, +15°, +30°, +45°, and +60°. In addition, there are two extreme pitch angles: −90° and +90°. We tested ViT models of different sizes. The main parameters to be adjusted are the latent dimension D and the number of self-attention heads H. We took 128, 256, 384 and 512 dimensions, corresponding to the 4, 4, 6 and 8 attention heads, respectively. The models in these three cases were tested and compared, and the results are shown in Table 3.

**Table 3.** Comparison of ViT parameters of different scales.

|      | Dimensions | Heads | Scale (Million) | Loss |
|------|-----------|-------|-----------------|------|
| VIT1 | 128 | 8 | 2.5 | 0.15 |
| VIT2 | 256 | 8 | 9.7 | 0.11 |
| VIT3 | 384 | 8 | 21.66 | 0.08 |
| VIT4 | 512 | 8 | 38.31 | 0.08 |

We set the training epoch to 200; the accuracy curve of the training set and verification set is shown in Figure 10. As the number of epochs increases, the training loss and the

verification loss decrease. When the epochs are greater than 150, the mean-square-loss decreases slowly. To prevent the model from over-fitting, we set the dropout rate to 0.15. Then, during model training, some hidden neuron nodes were set to be inactive with a 0.15 probability, and their adjacent links were removed. These nodes were ignored when the Adam optimizer optimizes, and the actual training parameters were reduced, which can prevent over-fitting.

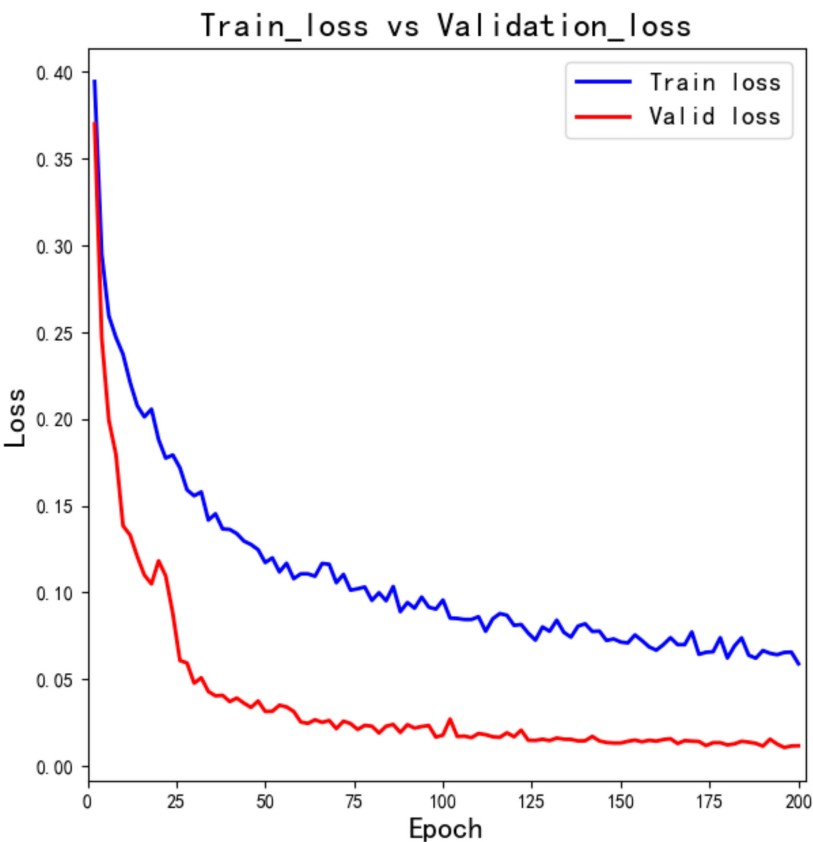

**Figure 10.** Training loss and validation loss of ViT.

With the increase in network scale, the network error becomes smaller, and the performance is better, but the training cycle increases, and it consumes more time and computing power. We made a trade-off between the network scale and performance and selected ViT3 in Table 3. In this paper, the ViT3 model of a 32 × 32 image patch is selected as the final model. Finally, we tested the network, and the average predicted loss was 8.6%. The predicted loss is shown in Table 4. The inference speed of the trained network is 5 FPS in a portable system and 48 FPS in an online workstation.

**Table 4.** Average loss of different head pose estimations.

|  | Left, −90 | Left, −60 | Left, t-30 | Front, 0 | Right, 30 | Right, 60 | Right, 90 |
|---|---|---|---|---|---|---|---|
| Bow, −60 | 0.075 | 0.077 | 0.069 | 0.072 | 0.085 | 0.074 | 0.085 |
| Bow, −30 | 0.082 | 0.103 | 0.092 | 0.088 | 0.058 | 0.088 | 0.086 |
| Straight, 0 | 0.094 | 0.081 | 0.087 | 0.096 | 0.098 | 0.088 | 0.095 |
| Raise, 30 | 0.084 | 0.097 | 0.099 | 0.086 | 0.111 | 0.081 | 0.087 |
| Raise, 60 | 0.088 | 0.079 | 0.092 | 0.092 | 0.08 | 0.077 | 0.069 |

### 3.2. Training and Use of Sound Classification Model

For the ambient sound classification model, we installed cameras and sound acquisition equipment in two classrooms. We used a CNN model for sound classification. Baidu's PaddleSpeech has a trained sound classification model [33], and the recognition rate reached more than 80%.

For the speech recognition model, we used PaddlePaddle trained end-to-end on the automatic speech recognition model PPASR [34], using CTC (connectionist temporal classification) loss as the loss function. It avoids the strict alignment operation required in the traditional speech recognition model. The direct input was complete speech data, and the output was the result of the whole sequence.

For the training and use of the voiceprint model, we used PaddlePaddle to achieve the voice recognition model [35]. The training data set used the Chinese corpus of voice corpus [36], which has 3242 voice data and 1,130,000+ voice data. The modified ResNet model was used in the model, and the input was the amplitude spectrum of a short-time Fourier transform. The experiment shows that the effect is acceptable.

### 3.3. Student Attention Analysis

We installed cameras and sound acquisition equipment in two classrooms. We used the CNN model for sound classification. Baidu's PaddleSpeech has a trained sound classification model with a recognition rate of more than 80% [37]. In these two classrooms, we collected data from two classes, with four courses and four teachers, totaling 40 class hours. Due to the impact of the COVID-19 pandemic, each class lasts 30 min. Owing to the slight noise in class, we optimized the program and added the volume feature so that the accuracy of class state classification can reach more than 90%. Table 5 shows 3 typical class state sequences, in which D denotes the lecturing state, I denotes the interactive state, P denotes the class practice state, and C denotes the transcription state.

**Table 5.** States in minutes of a typical class. The rows display the different classes, and the columns display the different 3-min states in minutes in class.

| Minute | 00–03 | 04–06 | 07–09 | 10–12 | 13–15 | 16–18 | 19–21 | 22–24 | 25–27 | 28–30 |
|--------|-------|-------|-------|-------|-------|-------|-------|-------|-------|-------|
| Les.1 | DDD | DCC | DDD | CCC | PPP | PPP | III | DDC | CCD | DDD |
| Les.2 | DDC | CCD | DDD | III | IDD | DDD | CCP | PPP | PPP | DDD |
| Les.3 | DDD | DDD | CCC | CCC | DDP | PPP | PPP | PPI | III | DDD |

The average learning attention of students is about 44.17%, of which the student learning attention is 25.23% of the lecturing state, 29.77% of the transcription state, 81.54% of the practice state, and 73.32% of the interaction state. Overall, the lecture state and transcription state account for about 64%, and the interaction and practice state account for about 36%, as shown in Table 6.

**Table 6.** Relationship between class state and learning attention.

|  | Duration Proportion (%) | Attention (%) |
|--|-------------------------|---------------|
| Lecturing | 41.23 | 25.23 |
| Transcription | 21.76 | 29.57 |
| Practice | 24.56 | 81.54 |
| Interaction | 12.45 | 73.32 |
| Average |  | 45.99 |

We also use the student's performance evaluation given by the teacher and final examination scores to test the students' learning attention in class to find if there exists

a correlation between students' performance evaluation (final examination scores) and students' learning attention. We found that a correlation exists. The students' performance evaluation and learning attention are more relevant, and the correlation between students' final examination results and learning attention is weaker but really exists. The result is shown in Table 7.

**Table 7.** Correlation between learning attention and performance evaluation (exam scores).

|  | Performance Evaluation of Students | Final Examination Score of Students |
|---|---|---|
| Learning attention of class one | 53.61% | 28.87% |
| Learning attention of class two | 52.63% | 48.28% |

We selected 12 students to sit in the front of the class, and we obtained their images in clear enough resolution to analyze their head poses. Normalized by histogram equalization, we rated the performance evaluation of each student into five grades, 100, 75, 50, 25, and 0, and did this same procedure to the examination scores. We also rated 12 students' learning attention into five grades, 100, 75, 50, 25, and 0, according to their average time of learning attention in class and by histogram equalization. Then, we calculated the correlation coefficient between students' leaning attention with their performance evaluation and the correlation between students' leaning attention with the examination score. The students' learning attention is both correlated with the performance evaluation and the final examination score. The above results show that it is effective to use this method to analyze students' attention. It can help the teacher to obtain his students' learning status.

*3.4. Comparison*

Many studies in the literature focused on the area of online learning; some online learning software can collect high-resolution videos of students' faces and consuct eye-gaze analysis, facial expression analysis, head pose analysis, etc., to accurately analyze students' learning attention. There are relatively few studies on learning attention analysis in offline class, for the camera is far away and contains each student in the image; additionally, the face is small, and the definition is inadequate. It is difficult to analyze the facial expression and eye states; only the head pose and body behaviors are easy to analyze. There are few existing papers that carry out their research by combining the learning attention with the class state. In fact, the students' attention corresponding to the same head pose is different in different class states. We combine voice and image information to form a multi-modal method that can improve the accuracy of learning attention recognition. Table 8 shows the comparison between our method and some existing methods. Because there is still a lack of a large-scale and publicly accepted learning attention dataset, it is difficult to conduct a fully quantitative comparative analysis on it.

**Table 8.** Comparison between different methods for learning attention recognition.

| Method | Attention Indicator | Environment (Online/Offline) | Key Techs. | Applied in Practice |
|---|---|---|---|---|
| Facial Expression Method [16] | Facial expression Eye state | Online | Gabor Wavelet SVM | Applied Online |
| Kinect Method [7] | Gaze, Body behavior | Offline | Dec. Trees K-NN | Unknown |
| eLearning Method [9] | Facial expression | Online | CNN SVM | Applied Online |
| Our Method | Head Pose Voice Class state | Offline | Retinaface ViT CNN | Applied in Offline classroom |

## 4. Conclusions

We found that there are many interactive processes between teachers and students in classroom teaching, students' learning attention is also high, and interaction has a higher degree of participation, which is the main reason for students' high attention. In the practice state of classroom teaching, students' attention is also relatively high because the goal of practice is highly oriented. To improve students' attention, we need to improve it from two aspects: students' internal factors and teachers' external teaching methods. Students should be given clear learning objectives; for example, adding some exercises in classroom teaching can also improve students' attention. As another example, giving students multiple small rewards in the learning process is an effective way to improve students' attention. In addition, increasing interactive learning content and carrying out project-based teaching can improve students' attention. In addition, we also found that in the state of independent seat selection, the students in the front row have high attention, and the students in the back row have low attention.

In this project, we found that occlusion is a big problem that prevents the system from detecting every face in class when the classroom is very big and there are many students in the classroom. The next step is to add multiple cameras to resolve the problem of occlusion of the back row students' face images, improve the precision of the back row face images, and collect the head video data with high definition to further improve the prediction accuracy of students' attention. The dataset we use to train and test our method in this paper is relatively small; we thus need to collect more snaps of different students and different classes to develop a larger dataset to train and test our method in the next study. The application of this method requires hardware and software systems to acquire and analyze audio and video data. Some schools have been equipped with the required system, but some have not. We can build low-cost portable equipment to meet these needs. In addition, with the development of AI technology, the cost of equipment will keep falling, and AI technology will continuously be applied to education, continuously helping improve education.

**Author Contributions:** Methodology, X.L., J.Y., J.L.; software, X.L., J.L.; validation, X.L., J.Y., J.L.; writing—original draft preparation, X.L., J.Y. and H.S.; writing—review and editing, X.L., J.Y. and H.Z.; visualization, X.L., J.Y. and H.Z. All authors have read and agreed to the published version of the manuscript.

**Funding:** This research was funded by The National Natural Science Foundation of China grant number 42075134 and the APC was funded by Shanghai Jiaotong University.

**Institutional Review Board Statement:** Ethical review and approval were waived for this study, due to the open data stated in Twitter's data privacy policy (https://twitter.com/en/privacy, accessed on 12 July 2022).

**Informed Consent Statement:** Informed consent was obtained from all subjects involved in the study.

**Data Availability Statement:** The data that support the findings of this study are available upon request to the corresponding author.

**Acknowledgments:** This research was supported by the National Natural Science Foundation of China (42075134). All authors have consented to the acknowledgement.

**Conflicts of Interest:** The authors declare no conflict of interest.

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
