# Peer review of "A Deep-Learning Based Method for Analysis of Students’ Attention in Offline Class"

_electronics, doi:10.3390/electronics11172663_

Round 1

Reviewer 1 Report (Previous Reviewer 3)

The authors noted in the response document that they have addressed the previous comments, but unfortunately, their revisions are insufficient.

1. It is unclear why the authors chose to change the title from "classroom" to "offline class"

2. We had pointed out several papers to include in their literature review. Unfortunately, they didn't go beyond those -- they only cited those papers. Moreover, they didn't discuss how their work is novel when others have already published papers with a similar focus.

3. The authors still didn't demonstrate the soundness of their study design. It is not enough to mention that "we ... collect and analyze the data of teachers, including ordinary teachers and star teachers ... different courses, some of which are relatively boring ... and some of which are interesting". It is essential to lay out the complete design of how teachers and courses were sampled and paired up before conducting the study. As a side note, designating teachers as "ordinary" and courses as "boring" seems to be an inappropriate and disrespectful attitude.

4. Comment on Figure 6: although it is not impossible, we rarely see training loss being greater than validation loss. Are the authors confident about this plot?

Author Response

The authors noted in the response document that they have addressed the previous comments, but unfortunately, their revisions are insufficient.

  1. It is unclear why the authors chose to change the title from "classroom" to "offline class".

Thank the worthy reviewer. The current research mainly focused in “Online class” environment. Researchers analyses student study attention by using online systems such Zoom Class, Tencent Class and so on. In “Online class” environment we can relatively easily acquire high resolution images of students faces, and the afterward research is also relatively easy.

Our research is focused on “traditional class” we called it “offline class”. In this kind of environment, we installed a camera in front of the classroom to take panoramic photos of all the student in the classroom. The face image of the students who are in the front of the classroom is big and clear, but the image of the students behind is small and unclear, and there also exist occlusion. So, our research in relatively more difficult, but it’s very important for a large majority of teaching is happened in traditional class currently.

  1. We had pointed out several papers to include in their literature review. Unfortunately, they didn't go beyond those -- they only cited those papers. Moreover, they didn't discuss how their work is novel when others have already published papers with a similar focus.

Thank the worthy reviewer for the worthy comments. We carefully studied the papers and find they are very helpful to our research! We have revised the manuscript and the changes are reflected at the page 2, line 79-83, page 3, line 90-103, color RED.

The novelty of our method is to use the new STATE-OF-ART deep learning network (such as Vision Transformer which is a better backbone neural network than CNN, RetinaFace which can detect human face precisely) and a multi-modal featured method (both voice model and computer vision model) to develop a system that has practical application to detect learning attention of students.

  1. The authors still didn't demonstrate the soundness of their study design. It is not enough to mention that "we ... collect and analyze the data of teachers, including ordinary teachers and star teachers ... different courses, some of which are relatively boring ... and some of which are interesting". It is essential to lay out the complete design of how teachers and courses were sampled and paired up beforeconducting the study. As a side note, designating teachers as "ordinary" and courses as "boring" seems to be an inappropriate and disrespectful attitude.

Thank the worthy reviewer for the worthy comments. Our main research is focus to apply the technologies of artificial intelligence into education area. We are not professional enough in the field of education. If our method is not perfect from the perspective of an expert in education area, we hope to we can get help from them in the future to improve our study. We will try to cooperation with educational expert to make our research better. We aim to use the latest deep-learning model in education area.

  • The authors still didn't demonstrate the soundness of their study design.

We are mainly engaged in empirical research in this paper, and we will use questionnaires to verify the results of results of the deeplearning method. We have revised the manuscript and the changes are reflected at the page 4, line 154-161, color RED.

(2) As a side note, designating teachers as "ordinary" and courses as "boring" seems to be an inappropriate and disrespectful attitude.

Thank the worthy reviewer for the worthy comments. We have revised the manuscript and the changes are reflected at the page 11, line 384-388, color RED.

  1. Comment on Figure 6: although it is not impossible, we rarely see training loss being greater than validation loss. Are the authors confident about this plot?

  We select a public dataset named “Head Pose Database” that all images have been taken by Prima Team of INRIA. The dataset is a small dataset. We have use data augmentation methods in training process to increase the training samples to train our model, but we do not need using data augmentation methods in validation process. In Pytorch, the data augmentation methods mainly are RandomResizedCrop, RandomHorizontalFlip, RandomRotation and RandomAffine. Since the data augmentation methods will cut a part of the training image or distort the training image, the training loss is larger than the validation loss. We are confident about the Figure 6 for we have observed this situation many times in our other training experiments.

  The Dataset is an open-source dataset. We can get the dataset by the URL:

https://gas.graviti.com/dataset/graviti/HeadPoseImage

Reviewer 2 Report (Previous Reviewer 2)

no comments, has reviewed previously 

Author Response

No modification

Reviewer 3 Report (Previous Reviewer 1)

The manuscript is revised significantly.

Author Response

No modification

Reviewer 4 Report (New Reviewer)

Dear Authors,

The following are the comments:

1. The work lacks sufficient validations to justify the adopted method. How can I justify the results on students' attention with an independent but authentic method? 

2. The physical posture can be completely irrelevant to the psychological factor. The psychological factor has been completely ignored, and only a superficial analysis has been presented.

3. Acquiring sufficient data and applying a DNN may be completely misleading. The authors should evaluate all the factors that affect the model and their correlations to have a proper insight. No effort has been taken in this regard.

4. There are several other aspects of facial features apart from the head pose that can play a significant role in the analysis of the student's attention

5. What's the training data? How was it acquired? How much authentic is the training data and was it verified independently?

6. Is this program deployable on an independent programmable hardware that runs on battery? If not its practical application will be limited. If yes, please provide the energy consumption analysis and the computational time.

7. There are numerous grammar errors throughout, please rectify

Author Response

  1. The work lacks sufficient validations to justify the adopted method. How can I justify the results on students' attention with an independent but authentic method? 

Thank the worthy reviewer for the worthy comments. We have revised the manuscript and the changes are reflected at the page 4, line 154-161, color RED.

We agree with the reviewer’s point, how to judge that a student is engaged in learning status is a complex task. This paper is aiming to use the deep learning techniques to develop a system to help teacher improving their teaching outcome. The proposed method in this paper is based on the following assumptions:

(1) When a teacher is giving a lecture, or interacts with students such as asking questions and waiting the students to answer, we think that students should look at the teacher. If the students are not look at the teacher, but lower their heads or faces to the left or right, then we don’t think they are focused.

(2) When the class is in the practice state, students should lower their heads and writing, otherwise we don’t think they are focused.

(3) When a student is in the state of taking notes, he should look up at the blackboard and then lower his head to copy. If he is not in this state, we also think he is not in the state of focus.

We will contact and cooperate with the experts in education to improve our research in the next.

  1. The physical posture can be completely irrelevant to the psychological factor. The psychological factor has been completely ignored, and only a superficial analysis has been presented.

In our research, we are mainly engaged in empirical research, and our model is based on the previous assumptions of student learning attention. Our method is helpful for teachers to improve their teaching skills.

Psychological factors are very important in our future research, may greatly improve our method. We hope to add more cameras in the back of the classroom to take high resolution images, then we can analyze facial expressions and eye gaze of each student, and further improve the effect of our model.

  1. Acquiring sufficient data and applying a DNN may be completely misleading. The authors should evaluate all the factors that affect the model and their correlations to have a proper insight. No effort has been taken in this regard.

We are researchers of artificial intelligence technology. We are committed to applying the latest state-of-art artificial intelligence technology into education area to help teachers to improve their teaching. We hope our efforts are in the right route. If our method deviates from education and psychology, we hope to we can get help of the experts in education in the future. We hope that in the follow-up work, we can continue to apply the latest artificial intelligence technology into education, in the right direction, liberating teachers from the tedious job such as observing, recording, collecting and appraisal for this kind of job can be well done by AI system. Then teachers can focus in their most valuable point, cultivating.

  1. There are several other aspects of facial features apart from the head pose that can play a significant role in the analysis of the student's attention.

We also observed that in addition to head pose, facial expressions, eye gaze status and other factors have an important impact on student’s learning attention. Then, for the limitation of the condition of the images collected by its camera in class, the faces of the students in the back of the classroom are very small, it is difficult to do eye gaze and facial expression analysis in the current condition. In the future, we will add more cameras, take high-resolution images to improve the research.

  1. What's the training data? How was it acquired? How much authentic is the training data and was it verified independently?

The changes are reflected from page 11, line 384-387.

(1) What’s the training data?

We select a public dataset named Head Pose Database as our dataset. All images have been taken using the FAME Platform of the PRIMA Team in INRIA Rhone-Alpes. The pitch angle ranges from - 90 ° to + 90 ° and the yaw angle ranges from - 60 ° to + 60 ° respectively, including 2790 images. For every person, 2 series of 93 images (93 different poses) are available. People in the database wear glasses or not and have a various skin color. The background is willingly neutral and uncluttered to focus on the face.

We divided the training data into 3 categorizes, training dataset, validation dataset and testing dataset, the sample ratio is 6:2:2.

(2) How was it acquired?

The Dataset is an open-source dataset. We can get the dataset by the URL:

 https://gas.graviti.com/dataset/graviti/HeadPoseImage

  • How much authentic is the training data and was it verified independently?

We have trained the head pose regression model in the year 2021, detail information can be find in our previous 2 papers:

A New Head Pose Estimation Method Using Vision Transformer Model

Xufeng Ling; Dong Wang; Jie Yang

2021 7th International Conference on Computing and Artificial Intelligence Proceedings

DOI 10.1145/3467707.3467729

A Facial Expression Recognition System for Smart Learning Based on YOLO and Vision Transformer

Xufeng Ling, Jingxin Liang, Dong Wang, Jie Yang

2021 7th International Conference on Computing and Artificial Intelligence Proceedings

DOI 10.1145/3467707.3467733

  1. Is this program deployable on an independent programmable hardware that runs on battery? If not its practical application will be limited. If yes, please provide the energy consumption analysis and the computational time.

This program can be deployed both on a NVIDIA-Xavier-based embedded system (total power is less than 200W) or a normal PC workstation (need to connect power supply). Since most classroom have power supply, in most time the system does not need an external battery. But we really provide NVIDIA-Xavier-based embedded system with an external battery named DXPOWER which has theoretically 1000WH when fully charged to meet outdoor usage.

        (photo1)            (photo2)

The left is the NVIDIA-Xavier system, and the right battery is DXPOWER. The DXPOWER can support the NVIDIA-Xavier-based embedded system for about more than 3.5 hours.

  1. There are numerous grammar errors throughout, please rectify.

We have correct some grammar errors. The changes are reflected in color Blue in page 1, 2, 4, 5, 6, 10,11,12,13,14,16.

Reviewer 5 Report (New Reviewer)

- In line 69 reference 11 written two times should be corrected. "Chang et al. [11][11] proposed"

Author Response

We resolve the issues of improvement highlighted by the worthy reviewer. The mistake has been fixed, page 2, line 71, color GREEN.

Round 2

Reviewer 1 Report (Previous Reviewer 3)

The study design is still a major weakness of this manuscript. Conclusions like "We found that the students’ average time of attention is higher when the star teachers teach" can only be made when both teachers were let to teach the same material and to the same set of students. 

It is understandable that when data is collected from a regular, ongoing school system, the above controlled experiment design is not always feasible. The problem is, this weakens the claims that one can make from this type of studies. The authors need to properly discuss this point and acknowledge it as a weakness; and they need to do this right from the Introduction.

Author Response

The worthy comments are very helpful to our research. We have try to adopted some pedagogical methods to testify the effectiveness of our research. We use the student's performance evaluation given by teacher and final examination scores to test the students' learning attention in class to find that if there exists correlation from students’ performance evaluation (final examination scores) with students' learning attention. We found that the correlation exists. The students' performance evaluation and learning attention are more relevant, and the correlation between students' final examination results and learning attention is weaker but really exists.

  1. The study design is still a major weakness of this manuscript. Conclusions like "We found that the students’ average time of attention is higher when the star teachers teach" can only be made when both teachers were let to teach the same material and to the same set of students. 

We have contacted two teachers last week and obtained the performance evaluation of each student in class, and we also got the final examination scores of the right course. we select 12 students for they sit in the front of the class and we can get their images clear enough to analyzing their head pose.

Normalized by histogram equalization, We rated the performance evaluation of each student into five grades, 100, 75, 50, 25, and 0, same job did to the examination scores. And we also rated 12 students’ learning attention into five grades, 100, 75, 50, 25, and 0 according to their average time of learning attention in class,also by histogram equalization. Then we calculated the correlation coefficient between students' leaning attention with the their comprehensive performance evaluation, and the correlation between students' leaning attention with the examination score.

And the revision is reflected in page 15, line 469-485. COLOR RED

  1. It is understandable that when data is collected from a regular, ongoing school system, the above controlled experiment design is not always feasible. The problem is, this weakens the claims that one can make from this type of studies. The authors need to properly discuss this point and acknowledge it as a weakness; and they need to do this right from the Introduction.

In our school, all classrooms are Installed with a camera in the front. Some of the cameras were installed several years ago, and the models are relatively old. The pictures taken are not clear enough, and it is difficult to obtain the students' facial expressions, head posture and body pose. The camera of the school is being updated at a rate of 20% per year, so after about 3 years, we can continuously collect high-definition images.

It is true that some schools do not have the conditions, but with the development of science and technology and the continuous decline of hardware prices, comprehensive data collection in the classroom can be achieved. In fact, the price of our portable camera acquisition system is also very low, and the cost is less than 500 US dollars.

And the revision is reflected in page 17, line 522-526. COLOR RED

Reviewer 4 Report (New Reviewer)

Dear Authors,

The work is still not satisfactory. The following are the comments:

1. For most of the earlier comments, the authors mentioned that they would do it as future work. This indicates that the current work is not sufficiently novel. The results have to be independently endorsed by an expert, else, how can the results be considered as correct? 

2. The models used for classification have to be shown pictorially.

3. Classifictaion results need a confusion matrix.

4. The data taken is an existing dataset, the models used also seem to be existing and no independent method has been used to justify the results. So, I cannot recommend the work

Author Response

Thank the worthy reviewer for the worthy comments and they are very helpful to our research. We have try to adopted some pedagogical methods to testify the effectiveness of our research. We use the student's performance evaluation given by teacher and final examination scores to test the students' learning attention in class to find that if there exists correlation from students’ performance evaluation (final examination scores) with students' learning attention. We found that the correlation exists. The students' performance evaluation and learning attention are more relevant, and the correlation between students' final examination results and learning attention is weaker but really exists.

  1. For most of the earlier comments, the authors mentioned that they would do it as future work. This indicates that the current work is not sufficiently novel. The results have to be independently endorsed by an expert, else, how can the results be considered as correct? 

We have contacted two teachers last week and obtained the performance evaluation of each student in class, and we also got the final examination scores of the right course. we select 12 students for they sit in the front of the class and we can get their images clear enough to analyzing their head pose.

Normalized by histogram equalization, We rated the performance evaluation of each student into five grades, 100, 75, 50, 25, and 0, same job did to the examination scores. And we also rated 12 students’ learning attention into five grades, 100, 75, 50, 25, and 0 according to their average time of learning attention in class,also by histogram equalization. Then we calculated the correlation coefficient between students' leaning attention with the their comprehensive performance evaluation, and the correlation between students' leaning attention with the examination score.

And the revision is reflected in page 15, line 469-485.

  1. The models used for classification have to be shown pictorially.

Thank the reviewer for the worthy comments. We add the illustration of model. The changes are reflected from page 9, line 272-274. Color BLUE.

  1. Classification results need a confusion matrix.

Our approach uses the ViT method to regress the two angles of the student's head. It may so we have no confusion matrix for it is a regression model. The subsequent classification is based on the explicit rules as follows:

If the upward angle of the head is greater than -15 degrees, forward-looking state.

If the downward angle of the head is greater than -15 degrees, head down state.

If the head side angle is greater than 20 degrees, head side state.

Shown in page 11, line 351.

  1. The data taken is an existing dataset, the models used also seem to be existing and no independent method has been used to justify the results. So, I cannot recommend the work

This model was developed by our team. We improved the ViT method and used linear regression to predict the two angles of head pose. We published the experiments result at the conference ICCAI 2021. Now we have improved the research and conducted more experiments. We hope that our research can be published in the journal to let more people know this method, to help education experts use the new model to obtain the head pose of the students more accurately .

Round 3

Reviewer 4 Report (New Reviewer)

Dear Authors,

1. It has been mentioned that a CNN model has been used for sound classification. What’s the structure of this model and classification results in the form of confusion matrix.

2. The plot of loss vs epoch is not clear and needs improvement 

Author Response

  1. It has been mentioned that a CNN model has been used for sound classification. What’s the structure of this model and classification results in the form of confusion matrix.

The worthy comments are very helpful to our research. We have used the ECAPA-TDNN model which introduced the SE (squeeze exercise) module and the channel attention mechanism to recognize human voices from the environmental sounds. This model won the champion in the international voiceprint recognition competition (VoxSRC2020). And Baidu's PaddleSpeech framework implemented this model and made it open source. ECAPA-TDNN is used to extract voiceprint features, and perform excellent in voiceprint recognition, the recognition error rate (EER) is as low as 0.95%.

We have used the Urbansound8k dataset to train the ECAPA-TDNN model. Urbansound8k is a public dataset widely used for automatic urban environmental sound classification research, including 10 category sounds: air conditioner, siren, children playing, dog bark, jackhammer, engine idling, gunshot, drilling, car horn, and street music. Data can be downloaded at the following address:  

https://zenodo.org/record/1203745/files/UrbanSound8K.tar.gz

We remove the children playing sound samples, and add human voice samples to form a new dataset to train our specific model.

We have added the structure of CNN model and add the classification confusion matrix as well, and the revision is reflected in page 11, line 326-346.

  1. The plot of loss vs epoch is not clear and needs improvement.

We have revised our manuscript according to the worthy comments and the revision is reflected in page 15, line 433-447

This manuscript is a resubmission of an earlier submission. The following is a list of the peer review reports and author responses from that submission.

Round 1

Reviewer 1 Report

1. The paper has limited contribution and novelty. There is a large scope available for improvement.

2. How sound data is merged with images captured in the classroom.

3. The manuscript need to address the issues of occulsion which is common in classroom environment. How to manage in the given setup.

4. Also, how background changes will be taken care which capturing the images.

5. There is no comparison with other state of the art methods to assess the performance of the proposed work.

6. Also, data set seems to be very small. Authors must work on making it diverse in nature by having snaps of multiple classes, different time to acquire images etc.

7. There are few text portion which can be removed to enhance the readability of the manuscript.

Reviewer 2 Report

The paper proposes a method to obtain and measure students attention in class by applying  deep-learning models and employing IoT, video, and audio processing techniques.

The paper topic and its research area is interested and vital in nowadays.  However, some important modifications should be taken into account:

1-    The author should clearly indicate that the proposed system is classified either as online or offline teaching system. This should be included in the abstract or even in paper title.

2-    The author stated that its contribution is in the employing of artificial intelligent in the analysis of student attention, and this is the difference when comparing it with traditional techniques. But, there are large number of works deal with this issue by using Deep learning.

3-    Many acronyms are presented firstly without its details such as in: row 92: CLBP,  row 145: ViT, ASR, …….etc.

4-    In row 125, th author stated that : "its is difficult to achieve such image accuracy ………..” but he hasn’t explained how deep learning achieve this in this paper?!!

5-    In row 175: two versions, portable and lightweight model are used. It needs justification.

6-    In the system flowchart shown in fig. 1 and afterward sections, one can see that the proposed system uses different techniques that considered as computationally expensive either in video or audio collection and processing of data respectively . However, no indication on the performance of time are presented. Neither on training nor in inferences?!!!

7-    Comparison??!!

Reviewer 3 Report

Ling et al. gathered data to analyze students' attention/focus during classroom in-person lectures. If properly designed and articulated, this study would be an important addition to the literature. However, we have major concerns about several aspects of the study design and presentation.

1. The authors did not cite and discuss several recent and relevant works published at reputable venues. Here is a short list.

https://ejmcm.com/article_2065.html

https://jivp-eurasipjournals.springeropen.com/articles/10.1186/s13640-017-0228-8

https://link.springer.com/article/10.1007/s10648-019-09514-z

https://iopscience.iop.org/article/10.1088/1755-1315/199/3/032042/pdf

2. Study design:

2A. In this type of study, the students should be randomly split into groups to ensure randomized control. There should be one group who attends the class lectures without knowing that there are equipments in the classroom collecting their data.

2B. Lecture design and teacher assignment should also be taken into the study design. Otherwise, one cannot make points on "interesting topic" and "star teacher".

3. Data presentation:

3A. Figure 1 is not clearly showing the details, e.g., of the steps discussed in section 2.4

3B. Table 1 and 2 are critical for this study but the authors do not justify these tables; they just mentioned "After a lot of experiments and analysis, according to experience, the relationship between these three states and head posture parameters is shown in Table 1." This type of statements are not acceptable.